# Engineering Gaussian states of light from a planar microcavity

**Mathias Van Regemortel[1*], Sylvain Ravets[2], Atac Imamoglu[2], Iacopo Carusotto[3] and Michiel Wouters[1]**

**1** TQC, Universiteit Antwerpen, Universiteitsplein 1, B-2610 Antwerpen, Belgium
**2** Institute of Quantum Electroncis, ETH Zürich, CH-8093 Zürich, Switzerland
**3** INO-CNR BEC Center and Dipartimento di Fisica, Università di Trento, via Sommarive 14, 38123 Povo, Italy

* Mathias.VanRegemortel@uantwerpen.be

## Abstract

Quantum fluids of light in a nonlinear planar microcavity can exhibit antibunched photon statistics at short distances due to repulsive polariton interactions. We show that, despite the weakness of the nonlinearity, the antibunching signal can be amplified orders of magnitude with an appropriate free-space optics scheme to select and interfere output modes. Our results are understood from the *unconventional photon blockade* perspective by analyzing the approximate Gaussian output state of the microcavity. In a second part, we illustrate how the temporal and spatial profile of the density-density correlation function of a fluid of light can be reconstructed with free-space optics. Also here the nontrivial (anti)bunching signal can be amplified significantly by shaping the light emitted by the microcavity.

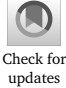
# 1   Introduction

Generation and manipulation of nonclassical states of light has been at the heart of quantum optics ever since its development in the early days [1]. A widely applied criterion to quantify the nonclassicality of a state is provided by the intensity correlations of the electromagnetic field $g^{(2)}(0) = \langle : \hat{I}^2 : \rangle / \langle \hat{I} \rangle^2$, with ':' denoting normal operator ordering. Whenever $g^{(2)}(0) < 1$, there is no classical analog of the quantum statistics and one is therefore led to conclude that the state has intrinsic quantum properties.

     The *photon blockade* is one of the most general mechanisms to realize these *antibunched* photon states. A strong nonlinear spectrum of the cavity makes it energetically forbidden for a second photon to enter once there is a first photon inside [2]. To date, a plethora of platforms exists in which this nonclassical feature of a light field has been demonstrated: by using a trapped atom in a cavity [3], with quantum dots [4,5] and, more recently, in superconducting circuits [6,7]. See Refs. [8–11] for recent reviews on these topics.

     Exciton-polaritons in planar microcavities, arising from the strong coupling between a cavity photon and a quantum-well exciton, provide a versatile platform to both generate and detect nontrivial photon features (see Refs. [12, 13] for recent reviews). The statistics of the polaritonic field is directly accessible through the photons that escape from the cavity, making these systems particularly interesting for measuring photon correlations. Moreover, the properties of the fluid, such as the density, velocity and phase, can be directly manipulated by the external laser field.

     In spite of these tremendous advantages, the major bottleneck of exciton-polariton systems has turned out to be the relatively weak interparticle interactions compared to currently achievable linewidths, ensuring that genuine quantum effects are often hidden by the strong incident laser field. Translated into the intensity fluctuations of the cavity output photon field, this means that $g^{(2)}(0) \approx 1$, the value for a coherent photon field. Nevertheless, albeit very small, there is a correction to this trivial value which stems from polariton-polariton interactions [14]. On the first level of approximation, this effect can be understood in terms of the creation of pairs of quantum fluctuations that propagate through the fluid in opposite direc-

tions. The statistics of these pairs of fluctuations is described in terms of a two-mode squeezed Gaussian state.

It was shown in a number of recent studies [15–18] that strongly antibunched photon statistics is not necessarily a consequence of a strong cavity nonlinearity. Certain setups consisting of two coupled cavities can exhibit a strong suppression of $g^{(2)}(0)$, despite having interactions that are substantially weaker than the typical dissipation rate. Dubbed the *unconventional photon blockade*, this remarkable effect was attributed to a destructive interference between two independent pathways for injecting a second photon in the cavity. In a later stage all these different ideas were somehow unified and clarified in the framework of optimally amplitude-squeezed Gaussian states [19]. Therefore, squeezed coherent states can display sizeable antibunching, provided the average value of the field and the squeezing are tuned in an optimal way [20]. The theoretically predicted antibunched signal was recently observed in an experiment with microwave photons on a superconducting circuit [21].

The crucial point here is the ability of the system to show a significant squeezing without the coherent part overwhelming the quantum fluctuations. In weakly interacting spatially extended systems, the main reason why quantum effects are weak is the dominant contribution of the coherent field over the fluctuations. For a spatially homogeneous system, however, the coherent field and quantum fluctuations are easily separated in Fourier space. Significant antibunching can then be achieved after a suitable attenuation of the $k = 0$ component of the light field. Previous proposals in the context of the *unconventional photon blockade* relied on a setup that consists of two coupled (0D) microcavities. The setup that we introduce in this work, illustrated in Fig. 1, utilizes the photons generated from quantum fluctuations in a 2D planar microcavity to engineer an interference between two squeezed intracavity modes, with opposite momentum $(\mathbf{k}, -\mathbf{k})$, and the coherent pumping field to produce the desired antibunched statistics.

The proposed setup consists of engineering an appropriate filtering scheme that exactly tailors single-mode states of the form discussed in Ref. [19]. With the introduction of a few linear optical elements, like beam splitters, phase shifters and attenuators, the optimal conditions for maximal suppression of $g^{(2)}(0)$ can be closely approached. We also compute the temporal profile $g^{(2)}(\tau)$ of delayed correlations to illustrate the expected sustain times of the antibunched signal, which is essentially limited by the photon lifetime.

While the cavity output field in the unconventional photon blockade is Gaussian within good approximation, it can still display highly nonclassical features due to the severely reduced intensity fluctuations. Consequently, the UPB provides an attractive mechanism for generating sizeable single photon sources, which can then be used as input for a computation scheme based on linear optics. However, the downside of the original proposal of two coupled cavities is the required fine tuning of system parameters like cavity coupling, photon nonlinearity and laser detuning [20]. In our scheme, this is in part circumvented by placing the interference and selection scheme *after* the microcavity, thus separating the squeezing and interference stage for increased control.

Since the antibunching in this setup originates from genuine particle interactions inside a planar microcavity, we devote the second part of this manuscript to investigating the spatial profile of correlations in the quantum fluid of light itself. By collecting and interfering all modes that escape from the microcavity, rather than isolating a single one, the real-space image of correlations can be reconstructed. Moreover, by carefully shaping the bundle of light before interference, it is possible to manipulate the (anti)bunched features in the spatial-temporal correlation pattern. Recently, the profile of density fluctuations has proven its importance in the context of analog gravity, as it encodes information about Hawking pair emission at a sonic hole horizon [22, 23], as was recently investigated in a cold-atom experiment [24].

The structure of our paper is as follows. We start by giving in Sec. 2 an overview of photon

correlations in quantum fluids of lights in ideal planar geometries. We then continue with explaining how to implement the unconventional photon blockade by filtering out specific modes in Sec. 3. Finally, in Sec. 4, we illustrate how the spatial-temporal profile of density-density correlations can be reconstructed with free-space optics. Technical details on the Bogoliubov approximation and on the time-dependence of operators are given in Appendices A and B, while Appendix C is devoted to a brief discussion of the principal imperfections and noise sources that may disturb the unconventional blockade effect in realistic planar microcavity devices.

## 2  Quantum fluctuations in fluids of light

In this section, we give an overview of the main features of the photon dynamics in nonlinear planar cavities under a coherent and monochromatic illumination. While our model is fully general, our discussion will be focused on the experimentally most relevant case of semiconductor microcavities with strong light-matter coupling, where the elementary luminous particles, the so-called exciton-polaritons, have a mixed light-matter character. For these systems, a wide theoretical and experimental literature has appeared that investigates the photon dynamics from the many-body perspective of the *quantum fluids of light* [12,13]. The interested reader should not find any difficulty in transferring our results to different material platforms and to different frequency domains.

We first set the stage by introducing the model that will be used in the following sections, we then review the dynamics of small collective excitations around the steady state as introduced in [25] and finally we summarize the main features of quantum correlations within a linearized approximation, as they were first presented in Ref. [26].

### 2.1  The model

We consider quasi-resonantly driven photons in a planar microcavity irradiated by a monochromatic and spatially plane-wave laser with frequency $\omega_L$ at normal incidence, as depicted in Fig. 1(a). After performing a unitary transformation to remove the time-dependence of the drive, the Hamiltonian in terms of the polariton field operator $\Psi(\mathbf{r})$, with $\mathbf{r} \equiv (x, y)$, is found as (we set $\hbar = 1$ throughout)

$$\hat{H} = \int d\mathbf{r}\left[ \hat{\Psi}^\dagger\left(\epsilon_{\mathrm{LP}}(-i\nabla) - \omega_L\right)\hat{\Psi} + \frac{g}{2}\hat{\Psi}^\dagger\hat{\Psi}^\dagger\hat{\Psi}\hat{\Psi} + F\left(\hat{\Psi} + \hat{\Psi}^\dagger\right)\right]. \tag{1}$$

The polaritonic modes have a dispersion $\epsilon_{\mathrm{LP}}(k)$ with a cut-off frequency at $\epsilon_{\mathrm{LP}}(0)$ and a low-energy behavior of the form $\epsilon_{\mathrm{LP}}(k) \approx \epsilon_{\mathrm{LP}}(0) + k^2/2m$ with an effective mass $m$. Photons are quasiresonantly injected into the cavity at a constant rate by the coherent laser, represented by the drive term with amplitude $F$.

The third-order optical nonlinearity is assumed to be defocusing and is described in the model as a contact potential with interaction constant $g > 0$. In the polariton case, this represents the repulsive interactions between quantum-well excitons. With typical values of the interaction constant of $g \sim 1 \mu eV \cdot \mu m^2$ and polariton masses of $m \sim 10^{-4} - 10^{-5} m_e$ (with $m_e$ the electron rest mass), we find that the dimensionless interaction constant $mg \approx 10^{-4} \ll 1$, so that the polariton system is clearly in the weakly interacting regime. This legitimates a linearized treatment of quantum fluctuations within a Bogoliubov approximation, as performed in the following.

Photons inside the cavity have a finite lifetime before they escape by tunneling through the cavity mirrors. The photonic losses can be described in the Born-Markov approximation

by introducing them into the master equation for the density matrix

$$\partial_t \hat{\rho} = -i[\hat{H}, \hat{\rho}] + \mathcal{D}[\hat{\rho}] \tag{2}$$

in the form of a dissipator of the Lindblad form

$$\mathcal{D}[\hat{\rho}] = \frac{\gamma}{2} \int d\mathbf{r} \left( 2\hat{\Psi}\hat{\rho}\hat{\Psi}^\dagger - \hat{\rho}\hat{n} - \hat{n}\hat{\rho} \right). \tag{3}$$

Here, $\tau = 1/\gamma$ is the polariton lifetime (typically of the order of $10 - 100$ ps) and $\hat{n} = \hat{\Psi}^\dagger \hat{\Psi}$.

Equivalently, the system dynamics can be formulated in terms of a *quantum Langevin equation* [27,28] for the polariton field operator $\hat{\Psi}(\mathbf{r}, t)$. In this framework, the equation of motion reads [12]

$$i\partial_t \hat{\Psi}(\mathbf{r}, t) = \left( -\frac{\nabla^2}{2m} - \delta + g\hat{\Psi}^\dagger(\mathbf{r}, t)\hat{\Psi}(\mathbf{r}, t) - i\frac{\gamma}{2} \right) \hat{\Psi}(\mathbf{r}, t) + F + \sqrt{\gamma}\hat{\xi}(\mathbf{r}, t), \tag{4}$$

where we have defined the detuning of the laser frequency from the bottom of the bare LP dispersion $\delta = \omega_L - \epsilon_{\mathrm{LP}}(0)$.

The non-unitary time evolution due to Markovian photon losses is represented by the imaginary loss term inside the brackets and by the noise operators $\hat{\xi}(\mathbf{r}, t)$, which assume Gaussian statistics. In the low-temperature regime ($k_B T \ll \omega_L$) under consideration here, their variance is given by

$$\left\langle \hat{\xi}(\mathbf{r}, t)\hat{\xi}(\mathbf{r}', t') \right\rangle = \left\langle \hat{\xi}^\dagger(\mathbf{r}, t)\hat{\xi}(\mathbf{r}', t') \right\rangle = 0, \tag{5}$$

$$\left\langle \hat{\xi}(\mathbf{r}, t)\hat{\xi}^\dagger(\mathbf{r}', t') \right\rangle = \delta(\mathbf{r} - \mathbf{r}')\delta(t - t'). \tag{6}$$

The quantum Langevin equation (4) will be the starting point of our analysis.

## 2.2 The linearized equations of motion

In the spatially uniform configuration considered here, we can conveniently parametrize the photon field operator as

$$\hat{\Psi}(\mathbf{r}, t) = \psi_0(t) + \hat{\phi}(\mathbf{r}, t) = \psi_0(t) + \frac{1}{\sqrt{V}} \sum_{\mathbf{k}} \hat{\phi}_{\mathbf{k}}(t)e^{i\mathbf{k}\cdot\mathbf{r}}, \tag{7}$$

where the classical field $\psi_0$ represents the coherent component and the quantum field $\hat{\phi}(\mathbf{r}, t)$ describes the fluctuations around it.

From a many-body perspective, the classical field $\psi_0$ can be seen as a condensate coherently created in the $\mathbf{k} = 0$ mode of the cavity by the incident laser beam. Owing to polariton-polariton interactions, the condensate is slightly depleted and a fraction of the condensate particles is converted into photons with nonzero momentum $\mathbf{k}$, as reflected by the fluctuation operators $\hat{\phi}_{\mathbf{k}}$. In the many-body language, this corresponds to the so-called quantum depletion of the condensate [29].

As a first step to solve the quantun Langevin equation (4), we can perform a mean-field approximation where the creation of these quantum fluctuations is neglected and only the condensate mode is assumed to be populated. In the steady state, the condensate density $n_0 = |\psi_0|^2$ can then be found as the solution of the polynomial equation

$$n_0\left(\Delta^2 + \gamma^2/4\right) = \left|F\right|^2, \tag{8}$$

with the interaction-renormalized detuning

$$\Delta = \delta - gn_0. \tag{9}$$

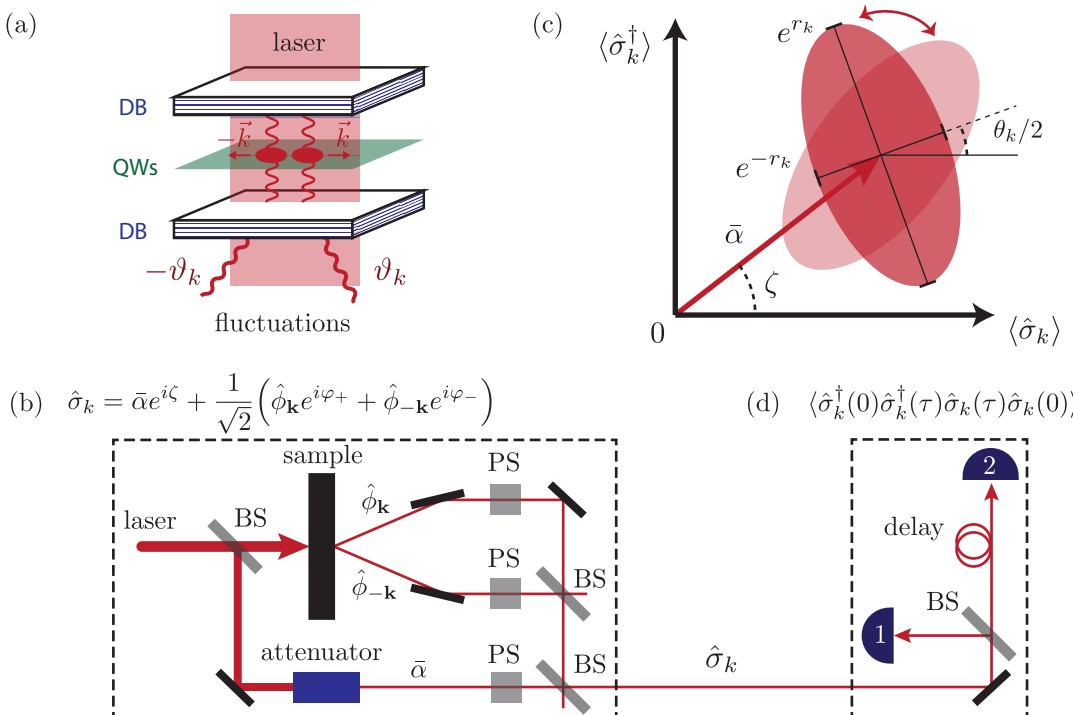

Figure 1: A schematic image of the selection and interference scheme that we propose. (a) A sketch of a planar semiconductor microcavity in the strong light-matter coupling regime. A quantum well (QW) is placed between two distributed Bragg mirrors (DB) and is pumped with a coherent laser beam. Photon interactions, mediated by the optical nonlinearity of the QW, lead to a small depletion of the condensate in terms of quantum fluctuations with nonzero momentum. The excitations are formed in pairs with opposite momentum and form a two-mode squeezed Gaussian state. A fluctuation with momentum **k** can leave the cavity as propagating radiation at an angle $\vartheta_k$, with $\sin\vartheta_k = ck/\omega_L$, from the perpendicular axis. (b) A schematic image of the setup, consisting of linear optical building blocks, that we propose to engineer squeezed coherent states as output. The coherent field of the pumping laser is attenuated and interfered with the two-mode squeezed Gaussian state of the quantum fluctuations to construct the photon field $\hat{\sigma}_k$. (c) The output state $\hat{\sigma}_k$ can be approximately parametrized as a squeezed coherent state. The squeezing parameter $r_k$ is momentum dependent and can be found through (29). The phase between squeezing ($\theta_k$) and displacement ($\zeta$) can be varied with the phase shifters from (b), as given in (30), allowing the output field to go from an amplitude to a phase squeezed state. (d) A HBT setup to measure photon correlations of the output state. Detecting simultaneous clicks of detectors 1 and 2 allows for the measurement of the delayed intensity fluctuations of the photon field $\hat{\sigma}_k$ and gives access to quantity (31). 'BS' stands for a (50:50) beam splitter and 'PS' for phase shifter.

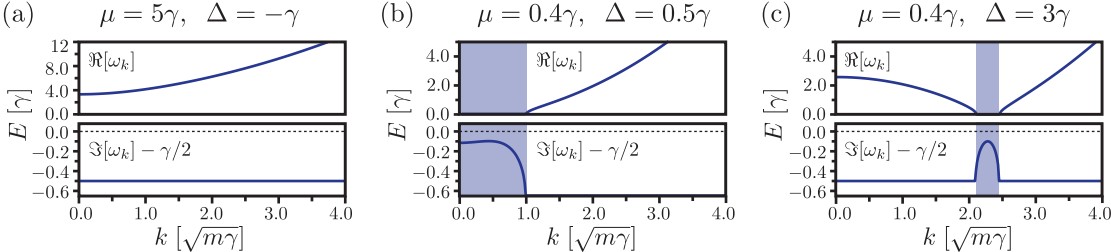

Figure 2: The quasiparticle spectrum from (12) for different mean-field configurations. (a) When $\Delta < 0$ the spectrum is gapped and there are no additional imaginary contributions. (b) When $0 < \Delta < 2\mu$, there is a disk of diffusive-like modes centred around $\mathbf{k} = 0$. These modes are characterized by having $\Re\omega_k = 0$, while having a nonzero imaginary contribution. (c) If $\Delta > 2\mu$, the diffusive modes shift to higher momentum and form a ring in momentum space. The diffusive regions are marked with blue shades. For clarity, we plot $\Im\omega_k - \gamma/2$, which represents the total decay rate of a Bogoliubov mode with momentum $\mathbf{k}$.

For given pumping conditions (defined by the detuning $\delta$ and the amplitude $F$), Equation (8) can have either one or two stable solutions for the mean-field density $n_0$. When $\delta < \sqrt{3}\gamma/2$ the resonator is in the optical-limiter regime. Alternatively, when $\delta > \sqrt{3}\gamma/2$ the system will exhibit a bistable behavior with a corresponding hysteresis curve [12]. For the analysis that follows, we will always implicitly determine $F$ by fixing a value of $n_0$.

Making use of the decomposition (7), an equation of motion for the fluctuation operators $\hat{\phi}_{\mathbf{k}}$ can then be derived out of the quantum Langevin equation (4). We will restrict to the lowest-order approximation of the interaction term in (4), which yields a set of linear equations for the zero-mean fluctuations $\hat{\phi}_{\mathbf{k}}$. Effects of higher-order interaction terms were recently studied in the context of Beliaev-Landau-type scatterings in the quantum fluid [30].

At our level of approximation, the motion of the linearized quantum fluctuations is determined by

$$i\partial_t \begin{pmatrix} \hat{\phi}_{\mathbf{k}} \\ \hat{\phi}^{\dagger}_{-\mathbf{k}} \end{pmatrix} = \left(B_{\mathbf{k}} - i\frac{\gamma}{2}\mathbb{I}\right)\begin{pmatrix} \hat{\phi}_{\mathbf{k}} \\ \hat{\phi}^{\dagger}_{-\mathbf{k}} \end{pmatrix} + \begin{pmatrix} \hat{\xi}_{\mathbf{k}} \\ \hat{\xi}^{\dagger}_{-\mathbf{k}} \end{pmatrix}, \quad B_{\mathbf{k}} = \begin{pmatrix} \varepsilon_{\mathbf{k}} + \mu & \mu \\ -\mu & -\varepsilon_{\mathbf{k}} - \mu \end{pmatrix}, \qquad (10)$$

where the Bogoliubov matrix $B_{\mathbf{k}}$ involves the usual interaction-induced off-diagonal coupling $\mu = gn_0$ but a shifted single-particle dispersion

$$\varepsilon_k = \frac{k^2}{2m} - \Delta = gn_0 + \epsilon_{LP}(0) + \frac{k^2}{2m} - \omega_L. \qquad (11)$$

The eigenvalues $\pm\omega_{\mathbf{k}}$ of $B_{\mathbf{k}}$ governing the dispersion relation of the linearized fluctuations have the same analytical form as the collective Bogoliubov excitations of a dilute Bose gas at equilibrium

$$\omega_k = \sqrt{\varepsilon_k(\varepsilon_k + 2\mu)}, \qquad (12)$$

but an important difference arises from the modified single-particle dispersion (11), which can either be gapped and strictly positive for $\Delta < 0$ or display regions of negative values for $\Delta > 0$.

Restricting to the most straightforward cases, a negative effective detuning $\Delta < 0$ is found in the $\delta < 0$ optical-limiter regime or on the high-density branch of the hysteresis loop in the $\delta > \sqrt{3}\gamma/2$ bistable regime. A positive effective detuning $\Delta > 0$ is instead found on the low-density branch of the hysteresis loop and is characterized by the Bogoliubov dispersion being purely imaginary in some regions, so that the dynamics of the elementary excitations is

*diffusive-like* (see Appendix A). Note that this regime is only parametrically stable when $2\mu < \gamma$, as we explain in Appendix B. In contrast to the equilibrium case where general arguments impose that the Bogoliubov dispersion is gapless [29], here the $\varepsilon_{k=0} = 0$ condition is only recovered in the special case $\Delta = 0$, i.e. at the end-point of the high-density branch. For clarification, we show the quasiparticle dispersion for different mean-field regimes in Fig. 2.

As a consequence of the photonic losses, the usual time-dependent Bogoliubov transformation, employed to diagonalize the equations of motion (10), acquires a damped exponential time dependence

$$\hat{\phi}_{\mathbf{k}}(t) = e^{-\gamma t/2}\Big(\eta_k(t)\hat{\phi}_{\mathbf{k}}(0) + \zeta_k(t)\hat{\phi}^{\dagger}_{-\mathbf{k}}(0)\Big) + \text{noise} \tag{13}$$

The expressions for the time-dependent Bogoliubov coefficients $\eta_k(t)$ and $\zeta_k(t)$ in the various mean-field regimes are presented in Appendix B. It should be noted, though, that the usual interpretation of the Bogoliubov operators as bosonic quasiparticles breaks down for the case of a diffusive dispersion. This is discussed in detail in Appendix A.

The freedom to tune $\Delta$ in the setup (by merely changing the pump frequency $\omega_L$) allows us to explore parameter regimes, inaccessible to the conservative dynamics of dilute Bose gases at equilibrium, that can lead to novel, exotic physics. Previous work in this context has addressed superfluidity features [25] and the related drag force of a driven-dissipative fluid flowing past a defect [31]. Remarkably, it was pointed out in [32] that the diffusive modes for $\Delta > 0$ can even give rise to a negative effective drag force.

## 2.3 Correlations in the steady state

From the stochastic equations of motion for the quantum fluctuations, presented in (10), we can derive equations for the evolution of the quadratic correlation functions. For this we define the momentum distribution $n_k = \langle \hat{\phi}^{\dagger}_{\mathbf{k}} \hat{\phi}_{\mathbf{k}} \rangle$ and the pair correlation $c_k = \langle \hat{\phi}_{\mathbf{k}} \hat{\phi}_{-\mathbf{k}} \rangle$ to find

$$\partial_t n_k = -\gamma n_k + 2\Im\big[g\psi_0^2 c_k^*\big] \tag{14}$$

$$i\partial_t c_k = (2\varepsilon_k + 2g|\psi_0|^2 - i\gamma)c_k + g\psi_0^2(2n_k + 1), \tag{15}$$

The steady state of these equations is readily evaluated by setting the left-hand side to zero and leads to the equal-time correlation functions in the stationary regime [26]

$$n_k = \frac{1}{2}\frac{(gn_0)^2}{\omega_k^2 + \gamma^2/4}, \quad c_k = -\frac{g\psi_0^2}{2}\frac{\varepsilon_k + gn_0 + i\gamma/2}{\omega_k^2 + \gamma^2/4}, \tag{16}$$

The nonequal-time correlations can also be found by making use of the Bogoliubov transformation (13). Thanks to the stationarity of the state, only the relative time difference $\tau = t - t'$ matters for the quantities $n_k(\tau) = \langle \hat{\phi}^{\dagger}_{\mathbf{k}}(t)\hat{\phi}_{\mathbf{k}}(t')\rangle$ and $c_k(\tau) = \langle \hat{\phi}_{\mathbf{k}}(t)\hat{\phi}_{-\mathbf{k}}(t')\rangle$. Based on the quantum regression theorem [27], the delayed correlation functions can be obtained by evolving the later operator over a time $\tau \geq 0$ with (13), which gives

$$n_k(\tau) = e^{-\gamma\tau/2}\Big(\eta_k^*(\tau)n_k + \zeta_k^*(\tau)c_k\Big), \quad c_k(\tau) = e^{-\gamma\tau/2}\Big(\eta_k(\tau)c_k + \zeta_k(\tau)n_k\Big), \tag{17}$$

in terms of the equal-time correlations $n_k$ and $c_k$ from (16).

# 3 Antibunched emission from squeezed quantum fluctuations

After introducing the general theoretical framework, we can start discussing how strongly antibunched light can be obtained by shaping the output of a weakly nonlinear coherently driven

planar microcavity. We start this section by reviewing the basic concepts of squeezed coherent states and derive theoretical bounds on the maximal amount of antibunching that can be obtained in the family of Gaussian states that we can engineer as output from the cavity. Following the lines of Ref. [19], we then propose a first example of a selection and interference scheme that is able to manipulate the output of a many-mode planar cavity and obtain antibunched light by letting three **k** modes symmetrically located at $\pm$**k** and **0** to mutually interfere. We finally characterize the intensity fluctuations and the level of antibunching that this scheme is able to produce.

## 3.1 Basics of Gaussian states

In the most general case, a single-mode squeezed state of the mode $\hat{a}$ is represented by the density matrix

$$\hat{\rho}_{\xi,n_{\text{th}}} = \hat{S}(\xi)\hat{\rho}_{n_{\text{th}}}\hat{S}^\dagger(\xi), \tag{18}$$

where $\hat{S}(\xi) = \exp\left[\frac{1}{2}(\xi^*\hat{a}^2 - \xi\hat{a}^{\dagger 2})\right]$ is the squeezing operator with $\xi = re^{i\theta}$. Here, $\rho_{n_{\text{th}}}$ is assumed to be a thermal density matrix with mean population $n_{\text{th}} = \text{tr}[\rho_{n_{\text{th}}}\hat{a}^\dagger\hat{a}]$.

Conversely, a Gaussian state is entirely characterized by its mean value and second moments and there exists a one-to-one map [1] that allows to extract squeezing parameters $\xi$ and $n_{\text{th}}$ from them,

$$n = \text{tr}[\hat{\rho}_{\xi,n_{\text{th}}}\hat{a}^\dagger\hat{a}] = \left(n_{\text{th}} + \frac{1}{2}\right)\cosh 2r - \frac{1}{2} \tag{19}$$

$$c = \text{tr}[\hat{\rho}_{\xi,n_{\text{th}}}\hat{a}\hat{a}] = -\left(n_{\text{th}} + \frac{1}{2}\right)e^{i\theta}\sinh 2r. \tag{20}$$

Additionally, the mode $\hat{a}$ can be displaced with a coherent field $\alpha = \bar{\alpha}e^{i\zeta}$, which leads to the new density matrix

$$\hat{\rho}_{\alpha,\xi,n_{\text{th}}} = \hat{D}(\alpha)\hat{\rho}_{\xi,n_{\text{th}}}\hat{D}^\dagger(\alpha), \tag{21}$$

where $\hat{D}(\alpha) = \exp[\alpha\hat{a}^\dagger - \alpha^*\hat{a}]$ is the displacement operator. The displacement field is then found back from $\hat{\rho}_{\alpha,\xi,n_{\text{th}}}$ by relating

$$\alpha = \text{tr}[\hat{\rho}_{\alpha,\xi,n_{\text{th}}}\hat{a}]. \tag{22}$$

A genuine signature of the non-classical nature of a state of light is provided by the intensity fluctuations of the photon field. The correlation function

$$g^{(2)}(0) = \frac{\langle : \hat{n}^2 : \rangle}{\langle \hat{n}\rangle^2} \tag{23}$$

with $\hat{n} = \hat{a}^\dagger\hat{a}$ can be shown to obey $g^{(2)}(0) \geq 1$ for any classical state of light. Consequently, a violation of this inequality is a manifest indication of quantum correlations in the photon state. It was shown in Ref. [19] how optimally amplitude-squeezed Gaussian states of type (21) can strongly violate the inequality. Specific relations were derived between displacement $\alpha$, squeezing $\xi$ and thermal density $n_{\text{th}}$ to attain the theoretical lower bound on $g^{(2)}(0)$.

In general, the equal-time second-order correlation function of a displaced, squeezed Gaussian state of form (21) reads

$$g^{(2)}(0) = 1 + \frac{2\bar{\alpha}^2(n - \bar{c}) + n^2 + \bar{c}^2}{\left(\bar{\alpha}^2 + n\right)^2} \tag{24}$$

Here $c = \bar{c}e^{i\theta}$ and we have set $\theta = 2\zeta$, so that the squeezing takes place exactly in the amplitude quadrature, thus obtaining optimal antibunching conditions.

When the second moments $n$ and $c$ of the fluctuations are fixed, we can vary the displacement field $\alpha$ to find optimal antibunching conditions from (24). After straightforward algebra we find that setting

$$\bar{\alpha}_{\mathrm{opt}} = \sqrt{\frac{(\bar{c}+n)\bar{c}}{\bar{c}-n}} \tag{25}$$

establishes optimal antibunching. The intensity fluctuations of this particular displacement field are then given by

$$g_{\mathrm{opt}}^{(2)}(0) = 1 - \frac{(\bar{c}-n)^2}{\bar{c}^2 + 2\bar{c}n - n^2}. \tag{26}$$

## 3.2 Engineering squeezed coherent output states

In the squeezing language of this section, the non-trivial quadratic correlations found in (16) reflect the fact that the two modes at $\pm\mathbf{k}$ are in a two-mode squeezed state. Similar to the single-mode case, the two-mode squeezing operator is defined as $\hat{S}_2(\xi) = \exp[-\xi\hat{a}\hat{b} + \xi^*\hat{a}^\dagger\hat{b}^\dagger]$, with in our case $\hat{a} \equiv \hat{\phi}_{\mathbf{k}}$ and $\hat{b} \equiv \hat{\phi}_{-\mathbf{k}}$. The main idea of this section is to illustrate that the intrinsic squeezing of the quantum fluctuations in the oppositely propagating modes can be utilized to construct a strongly antibunched output beam.

Our proposal is based on the fact that the emission angle $\vartheta_k$ of photons escaping a planar cavity is directly related to their cavity in-plane momentum $\mathbf{k}$ via $\sin\vartheta_k = ck/\omega_{\mathrm{L}}$, where $c$ is the speed of light in vacuum and $\omega_{\mathrm{L}}$ is the laser angular frequency. Quantum fluctuations with in-plane momentum $k$ will therefore lead to emission of pairs of photons at angles $\pm\vartheta_k$, which can conveniently be isolated and later interfered to obtain a single-mode squeezed state as show in Fig. 1a). We illustrate in a schematic way (see Fig. 1b), the setup we propose to achieve this goal. We create a coherent population of polaritons by shining laser light onto the sample. The laser is tuned to the $\mathbf{k} = 0$ lower polariton energy so as to only excite resonantly $\mathbf{k} \simeq 0$ polaritons. Quantum fluctuations with momentum $k > 0$ will lead to emission of pairs of photons at angles $\pm\vartheta_k$. The proposed experiment consists of combining the photons originating from the $\pm\mathbf{k}$ modes onto a 50:50 beam splitter. One simple way of realizing this interference uses lenses to image the Fourier plane on two pinholes (or equivalently on the cores of two single-mode fibers) that only transmit the desired Fourier components, and to recombine them on a free-space (or fiber) beam splitter. We also suggest that imaging the Fourier space directly on an optical fiber bundle or on a spatial light modulator that selectively transmits the chosen modes would provide a more compact and elegant experimental implementation of this interference. After recombining these two modes, the light is interfered with a coherent field obtained by suitably attenuating and phase-shifting the pumping laser. Photon correlations are finally measured in a standard Hanbury Brown and Twiss (HBT) setup.

The final output state after selection and interference is then found to be of the form

$$\hat{\sigma}_k = \bar{\alpha}e^{i\zeta} + \frac{1}{\sqrt{2}}\left(\hat{\phi}_{\mathbf{k}}e^{i\varphi_+} + \hat{\phi}_{-\mathbf{k}}e^{i\varphi_-}\right), \tag{27}$$

with $\zeta$ and $\varphi_\pm$ the accumulated relative phases in the arms. By evaluating its expectation value $\bar{\alpha}e^{i\zeta}$ and its quadratic correlation functions,

$$\langle\hat{\sigma}_k^\dagger\hat{\sigma}_k\rangle = \bar{\alpha}^2 + n_k, \quad \langle\hat{\sigma}_k\hat{\sigma}_k\rangle = \bar{\alpha}^2 e^{2i\zeta} + c_k e^{i(\varphi_+ + \varphi_-)}, \tag{28}$$

where $n_k$ and $c_k$ are given in (16), it is straightforward to see that the mode $\hat{\sigma}_k$ can be regarded as a squeezed coherent state of form (21). Thanks to the $\mathbf{k} \to -\mathbf{k}$ symmetry of the setup, the two-mode nature of the squeezing results here in exactly the same statistics as expected for a single-mode squeezed state.

We can next determine the effective squeezing parameter $r_k e^{i\theta_k}$ and thermal density $n_{\text{th},k}$ for the output mode $\hat{\sigma}_k$. After straightforward algebra, we find from expressions (19, 20) the thermal density and squeezing of output state $\hat{\sigma}_k$ (27)

$$n_{\text{th},k} = \sqrt{\left(n_k + \tfrac{1}{2}\right)^2 + |c_k|^2} - \tfrac{1}{2}, \quad \tanh 2r_k = \frac{|c_k|}{n_k + \tfrac{1}{2}}. \tag{29}$$

with $n_k$ and $c_k$ found within the Bogoliubov framework and given in (16). Additionally, we see that the squeezing phase is found as

$$\theta_k = \arg c_k + \varphi_+ + \varphi_-. \tag{30}$$

Therefore, the difference between $\theta_k$ and the displacement phase $\zeta$ can be tuned by varying $\varphi_+$, $\varphi_-$ and $\zeta$ with the phase shifters in the setup from Fig. 1b). In the schematic image of the squeezed state shown in Fig. 1c), this corresponds to rotating the ellipse, which allows to switch between an amplitude and a phase squeezed state.

In Fig. 3(a-b) we show how the parameters $r_k$ and $n_{\text{th},k}$ from expression (29) depend on the selected momentum **k** in the two cases of respectively negative (Fig. 3b) and positive (Fig. 3a) values of the interaction-renormalized detuning $\Delta$ defined in (9). Since excitations become generally more particle-like and have a larger frequency at large $k$, we expect the squeezing parameter $r_k$ to drop to zero in this limit. However, also their number $n_{\text{th},k}$ goes to zero in the same limit as it is generally less likely to excite fluctuations with higher momenta. We can anticipate at this point that the decay of $n_{\text{th},k}$ leads, in principle, to an asymptotic perfect antibunching of the photon statistics in the output state.

While both $n_{\text{th}}$ and $r$ monotonously drop to zero in the $\Delta < 0$ case (as can be observed in Fig. 3b), the presence of a set of diffusive modes in the $\Delta > 0$ case from Fig. 3a (see Appendix B) leads to a more versatile behavior. As is reflected by the peak in $n_{\text{th}}$ at nonzero momentum, these diffusive modes are parametrically amplified. In addition, these modes are also strongly squeezed, which we conclude from the enhanced squeezing parameter $r$ for the same momentum values as the peak in $n_{\text{th}}$.

### 3.3 Optimizing the antibunching

The freedom to vary at will the attenuation level and the phase shift experienced by the coherent laser field in the setup from Fig. 1b) permits to approach the condition (25) for optimal antibunching. A measurement of the temporal correlation of the intensity fluctuations of the output state $\hat{\sigma}_k$ (27) gives access to the quantity

$$g^{(2)}(\tau) = \frac{\langle \hat{\sigma}_k^{\dagger}(0)\hat{\sigma}_k^{\dagger}(\tau)\hat{\sigma}_k(\tau)\hat{\sigma}_k(0)\rangle}{\langle \hat{\sigma}_k^{\dagger}\hat{\sigma}_k\rangle^2} = 1 + \frac{2\bar{\alpha}^2 \Re\left\{n_k(\tau) + c_k(\tau)e^{i\eta}\right\} + n_k^2(\tau) + c_k^2(\tau)}{\left(\bar{\alpha}^2 + n_k(0)\right)^2}, \tag{31}$$

where $n_k(\tau)$ and $c_k(\tau)$ are given in (17) and the total phase $\eta = \varphi_+ + \varphi_- - 2\zeta$. In Fig. 1d we provide a schematic image of a Hanbury-Brown-Twist (HBT) setup to illustrate how this quantity is typically measured.

Let us first start by analyzing $g^{(2)}(0)$ (i.e at zero time delay) for the case where the phases are tuned such that squeezing is exactly in the amplitude quadrature, as realized by setting

$$\eta \to \eta_{\text{opt}} = \pi - \arg c_k(0). \tag{32}$$

This condition amounts to rotating the major axis of the ellipse in Fig. 1c) into the direction perpendicular to the displacement vector.

In Fig. 3(c,d) we show how the optimal displacement field $\alpha_{\text{opt}}$ (see (25)) and the minimal value of $g^{(2)}(0)$ (see (26)) depend upon the selected momentum **k** in the setup from Fig. 1b).

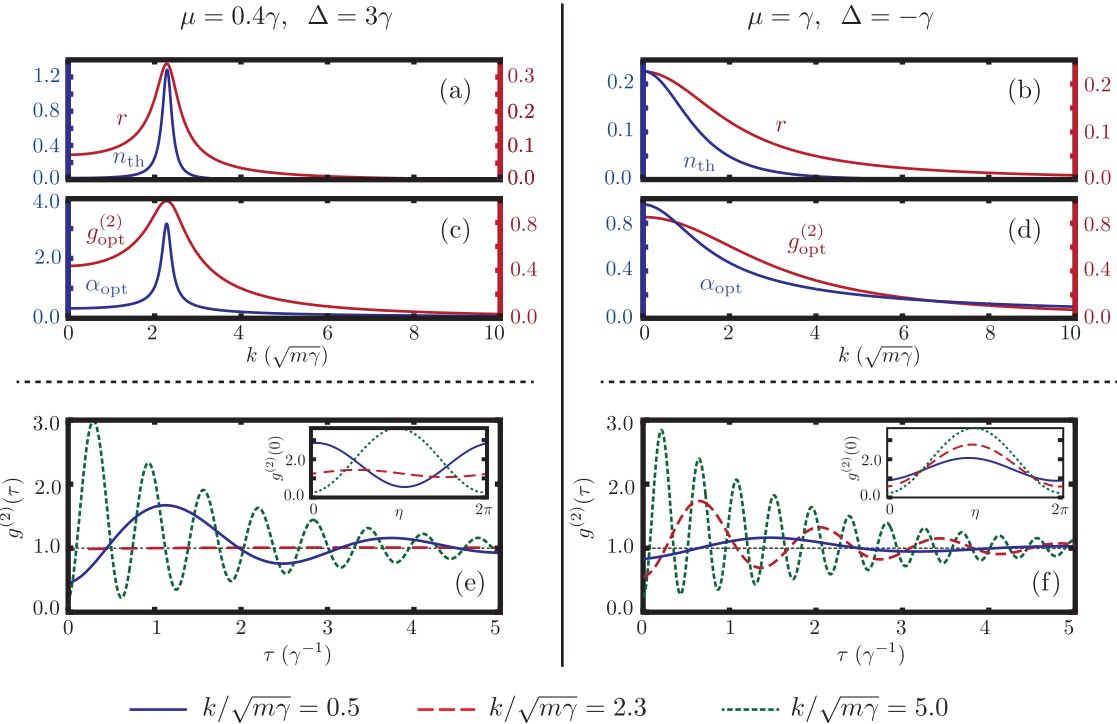

Figure 3: (a,b) The squeezing parameter $r$ and the thermal occupation $n_{\text{th}}$ as a function of momentum $k$ for a steady state with a positive (a) and a negative (b) interaction-renormalized detuning (8). (c,d) The optimal displacement amplitude $\alpha_{\text{opt}}$ (21) and corresponding $g_{\text{opt}}^{(2)} = g^{(2)}(0)|_{\text{min}}$ (26) as a function of momentum for the same parameters as above. (e,f) The temporal profile of $g^{(2)}(\tau)$ after selecting various momenta as indicated below. The displacement amplitude $\bar{\alpha}$ and phase $\eta$ have been chosen to fulfil the optimal antibunching conditions (as derived from (c,d)) at $\tau = 0$. The insets show $g^{(2)}(0)$ upon varying the total phase between squeezing and displacement (corresponding to rotating the ellipse in Fig. 1c)) for the showed momenta.

At high momenta $g^{(2)}(0)|_{\text{min}}$ always drops to zero, meaning that we can, in principle, approach a perfect antibunching by selecting higher momenta. This can be understood by noticing that $g^{(2)}(0)|_{\text{min}} \approx 8\sqrt{n_{\text{th}}}$ for small $n_{\text{th}}$ in the case of optimal squeezing and displacement, as was explained in Ref. [19]. While we have experimental control to tune $\alpha$ to its optimal value, the squeezing parameter $r$ is set by the nature of the nonlinear processes inside the cavity. We can verify that $r \neq r_{\text{opt}}$ in general, but, interestingly, we still find that $g^{(2)}(0)|_{\text{min}} \to 0$ in the limit $n_{\text{th}} \to 0$. Note also that, unfortunately, $\alpha_{\text{opt}}$ approaches zero in this limit as well, such that the total photon flux is expected to become vanishingly small.

For intermediate values of $k$, the behavior is different according to the sign of $\Delta$. While in the $\Delta < 0$ case, a monotonously decaying behavior is observed in Fig 3d) for both $\alpha_{\text{opt}}$ and $g^{(2)}(0)|_{\text{min}}$ as a function of selected momentum $\mathbf{k}$, the presence of parametrically amplified modes results in a strong increase of $n_{\text{th},k}$ and $r_k$ in the $\Delta > 0$ case, shown in Fig 3c) . The overall effect on $g^{(2)}(0)|_{\text{min}}$ is, however, detrimental, as this quantity is pushed back towards its value $g^{2)}(0)|_{\text{min}} \approx 1$ for a classical coherent field.

Let us now move to the full temporal dependence of $g^{(2)}(\tau)$, as expressed in (31). In Fig. 3(e-f) we analyze the delayed second-order correlations $g^{(2)}(\tau)$ for the displacement amplitude $\tilde{\alpha} = \tilde{\alpha}_{\text{opt}}$ that optimizes $g^{(2)}(0)$ (see expression (25)) for different values of $k/\sqrt{m\gamma}$.

In general we see from Fig. 3(e,f) that $g^{(2)}(\tau)$ shows a damped oscillatory behavior with an oscillation frequency set by $\Re\omega_k$ (i.e. the Bogoliubov frequency of the selected Fourier component), and a damping equal to $\gamma/2 - \Im\omega_k$ (i.e. the lifetime of the associated Bogoliubov mode). Moreover, we can shift the offset of the oscillation by varying the phase $\eta$, which we illustrate in the insets of Fig. 3(e-f). In the main panels (a,b), the offset of $g^{(2)}(\tau)$ has been chosen such that maximal antibunching is achieved at $\tau = 0$, corresponding to setting $\eta = \eta_{\text{opt}}$ from (32) to realize the result presented in Fig. 3(c,d). The opposite choice of the relative phase would instead give the typical enhanced intensity fluctuations of a phase-squeezed state.

As already mentioned, a stronger $\tau = 0$ antibunching can in principle be obtained by selecting higher momentum modes with the setup from Fig. 1b), but in Fig.3(e,f) we see that the oscillation frequency increases as well: the rapid fluctuations of $g^{(2)}(\tau)$ between low and high values mean that the initial antibunching signal can be easily washed away by the finite response time of a realistic detector, typically on the order or even longer than the photon lifetime in the cavity.

Another experimental difficulty may arise from the requirement of a spatially homogeneous fluid inside the microcavity. Disorder along the cavity plane may lead to unwanted scattering that could spoil the signatures of the coherent pair-creation processes, as we investigate in Appendix C.1. Additionally, the interaction with a thermal phonon bath may lead to dephasing and a reduced squeezing of the output photons; this is discussed in Appendix C.2. Finally, various sorts of uncontrolled relaxation mechanisms could lead to the building up of a thermal polariton population, see Appendix C.3.

## 4 Manipulating and probing the photon statistics with a wide-aperture lens

In the previous section we have introduced a first example of an optical interference scheme that is able to convert the (weak) squeezing of in-cavity modes into a (sizeably) antibunched single-mode output beam. The proposed setup required the isolation of three **k** components and the subsequent manipulation by a series of linear optical elements.

The present section exploits in-cavity interference between all **k** modes in real space to enhance the antibunching statistics of the output beam while keeping its qualitative spatial profile. The proposed configuration is depicted in Fig. 4a): A spatial image of the cavity field can be reconstructed by placing a system of two wide-aperture lenses in a confocal configuration in front of the microcavity. In the focal plane between the two lenses, a space-dependent attenuation element, e.g. based on a Spatial Light Modulator (SLM) sketched in Fig. 4b), provides the required **k** selection mechanism that is necessary to reduce the amplitude of the **k** = 0 mode with a factor $\mathcal{F}$, while keeping other **k** components intact.

A most remarkable feature of this alternative scheme is that the antibunching statistics results here from the sum of contributions of all **k** modes. In the following, we will illustrate how the present setup, providing the freedom to vary the coherent amplitude, offers a wide flexibility in the design of the spatio-temporal pattern of photon statistics.

### 4.1 Intensity correlations in position space

From ansatz (7), it is immediate to see that the real-space two-point correlation functions of fluctuations in a spatially uniform setup can be obtained from (17) as the Fourier transforms

of the quantities $n_k(\tau)$ and $c_k(\tau)$

$$n(x,\tau) \;=\; \left\langle \hat{\phi}^\dagger(\mathbf{r},t)\hat{\phi}(\mathbf{r}',t')\right\rangle = \frac{1}{V}\sum_{\mathbf{k}} n_k(\tau)e^{i\mathbf{k}\cdot(\mathbf{r}-\mathbf{r}')}, \tag{33}$$

$$c(x,\tau) \;=\; \left\langle \hat{\phi}(\mathbf{r},t)\hat{\phi}(\mathbf{r}',t')\right\rangle = \frac{1}{V}\sum_{\mathbf{k}} c_k(\tau)e^{i\mathbf{k}\cdot(\mathbf{r}-\mathbf{r}')}, \tag{34}$$

with $x = |\mathbf{r}-\mathbf{r}'|$, reflecting the homogeneity of the setup.

From equations (16), one sees that at large $k$ the quadratic correlation functions behave as $n_k \sim k^{-4}$ and $c_k \sim k^{-2}$, which are universal scaling laws for gases interacting with a contact potential [29]. While this poses no issues for the first-order correlation function $n(x,\tau)$, the pair correlation $c(x,\tau)$ suffers from an ultraviolet divergence in two or more spatial dimensions in the limit of vanishing separation $x$. In other words, the convenient introduction of a zero-ranged contact interaction has the inconvenient consequence that pair correlations are only reproduced correctly for separations $x$ much larger than the true range of the interaction potential. Interpolation between the two-body problem with the correct scattering potential at short distances and the many-body result at large distances has proven to be a way out of this issue [33]. In our specific case, the unavoidable finite aperture of all optical elements naturally imposes an ultraviolet cut-off in $k$.

## 4.2 The density-density correlation function

In general, the density-density correlation function at nonzero spatial separation and time delay is equal to

$$g^{(2)}(\mathbf{r},t;\mathbf{r}',t') = \frac{\left\langle \hat{\Psi}^\dagger(\mathbf{r},t)\hat{\Psi}^\dagger(\mathbf{r}',t')\hat{\Psi}(\mathbf{r}',t')\hat{\Psi}(\mathbf{r},t)\right\rangle}{\left\langle \hat{\Psi}^\dagger(\mathbf{r},t)\hat{\Psi}(\mathbf{r},t)\right\rangle\left\langle \hat{\Psi}^\dagger(\mathbf{r}',t')\hat{\Psi}(\mathbf{r}',t')\right\rangle}. \tag{35}$$

Under the assumption of Gaussian fluctuations for the photon field, the density correlation function can be expressed by Wick theorem in terms of the two-point correlation functions and a coherent displacement field,

$$g^{(2)}(x,\tau) = 1 + \frac{\Re\left(|\psi_f|^2 n(x,\tau) + \psi_f^{*2} c(x,\tau)\right) + |n(x,\tau)|^2 + |c(x,\tau)|^2}{(|\psi_f|^2 + \delta n)^2}, \tag{36}$$

where $\delta n = n(0,0)$ is the density of noncondensed particles and $\psi_f = \mathcal{F}\psi_0$ (see Expr. 7) is the attenuated condensate mode.

From expression (36) one sees that the density correlation function suffers from the same ultraviolet divergence as the pair-correlations when a zero-ranged contact interaction is used. In our discussion, we therefore focus on nonzero separations $x$, significantly larger than the true potential range, such that our results do not suffer from this issue. In practice, we see that for each non-zero point $(x,\tau)$, the spatial-temporal profile of $g^{(2)}(x,\tau)$ converges to a well-defined value for a sufficiently large cutoff and solely the point $(0,0)$ is suffers from the ultraviolet divergence. For this analysis, we choose the cutoff high enough such that the profiles are converged. In any practical experiment, a cutoff in momentum space is always introduced by the finite aperture of the lenses used in the imaging system.

The analysis of our numerical calculation for $g^{(2)}(x,\tau)$ as a function of $x$ and $\tau$ is discussed in the next sections for the physically most remarkable cases. We will show the results for a polariton system with dimensionless interaction constant $mg = 10^{-4}$, but we emphasize that our results stand regardless of the value of the interaction constant $g$. A lower $g$ merely requires a stronger attenuation $\mathcal{F}$ of the $k = 0$ mode, provided the mean-field interaction energy $\mu = gn_0$, with $n_0$ the density of particles in the condensate, remains invariant.

## 4.3 Lightcone-like correlations in the high-density regime

We first analyze the $\Delta < 0$ case, which is obtained in the optical-limiter regime under the additional condition $\mu > \gamma, |\Delta|$. Except for the energy gap in the very low-$k$ range, the quasiparticles that arise in this case share similar features with the familiar phononic modes in an equilibrium condensate because interactions (quantified by $\mu$) dominate over losses (quantified by $\gamma$). We therefore expect that the elementary excitations, once created, can travel through the fluid at the speed of sound during a time roughly corresponding to the average photon lifetime $\tau = \gamma^{-1}$. The spatial-temporal profile of the density-density fluctuations is then expected to exhibit features that we can relate to a lightcone-like propagation of the quasiparticles. In Fig. 4d) we show the profiles of a fluid pumped with $\mu = 5\gamma$ and $\Delta = -\gamma$ for different attenuation levels $\mathcal{F}$ of the $\mathbf{k} \approx 0$ modes, which give different amplitudes of the displacement fields $\psi_f$.

It is illuminating to first discuss the case with $\mathcal{F} = 0$, i.e the situation with a completely attenuated coherent field. In that case we are only looking at the photon statistics that originate from the quantum fluctuations, without interference of the coherent field. As the quasiparticles are created in pairs with opposite momenta, we expect to observe bunched photon statistics in the spatial-temporal profile of $g^{(2)}(x, \tau)$. We see in Fig. 4d that for $\mathcal{F} = 0$ (upper panel) all statistics in the profile exhibits bunching, with higher values at $(x, \tau)$-points that can be related to creation processes of quasiparticle pairs. Most notably, an oscillation with a frequency $|\Delta|$ in time is seen, which is the frequency of low-momentum quasiparticles.

By varying the attenuation $\mathcal{F}$ of the $k = 0$ mode with the SLM (see Fig. 4b)) to add a coherent field $\psi_f$ to the signal, we can drastically change the appearance of the spatial-temporal profile of $g^{(2)}(x, \tau)$. Upon increasing $\mathcal{F}$ (i.e. transmitting a fraction of the condensate mode with the SLM, rather than blocking everything) we observe how the bunching of the quasiparticle pairs turns into antibunching in a sound-like band due to interference with the condensate mode (see the middle panel with $\mathcal{F} = 0.009$). For a larger fraction of $\psi_f$ (see panel in (d) with $\mathcal{F} = 0.018$) the profile of $g^{(2)}(x, \tau)$ remains largely the same in shape, but the variation from $g^{(2)} = 1$, the statistics of a fully coherent state, reduces.

The expected antibunched statistics at short times and distances, stemming from interparticle repulsion, is apparent for sufficiently high displacement fields $\psi_f$ (including the standard case of no attenuation). At later times we see how the same-point antibunching quickly disappears on a time-scale of the order of a fraction of $1/\mu$ and transforms into a propagating antibunching feature that travels through the fluid (see blue band) at a velocity close to $2c$, with $c = \sqrt{\mu/m}$ being the speed of sound (white dashed lines).

Taking inspiration from well-known results on quantum quenches and correlation functions in conservative systems [34–37], a (somewhat hand-waving) physical interpretation of this result is the following. Detection of a photon at the initial time and position creates a Bogoliubov quasiparticle in the photon fluid, which then travels away with the group velocity of the mode it is emitted into. In turn, the presence of this Bogoliubov quasiparticle modifies the probability of detecting a second photon at space-time points located along its world-line. Correlations are peaked on the lightcone, but the fact that they display a sizeable intensity both inside and outside the lightcones is due to the strong $k$-dependence of the Bogoliubov group velocity, that increases in a faster-than-linear way at large $k$ because of the quadratic cavity dispersion, and tends to zero for small $k$ because of the energy gap characteristic of the driven-dissipative condition.

## 4.4 Spatial pattern formation with diffusive modes

When the fluid is pumped on the lower-intensity branch of a bistable regime, a range of diffusive-like Bogoliubov modes is present. Provided losses still dominate over interactions

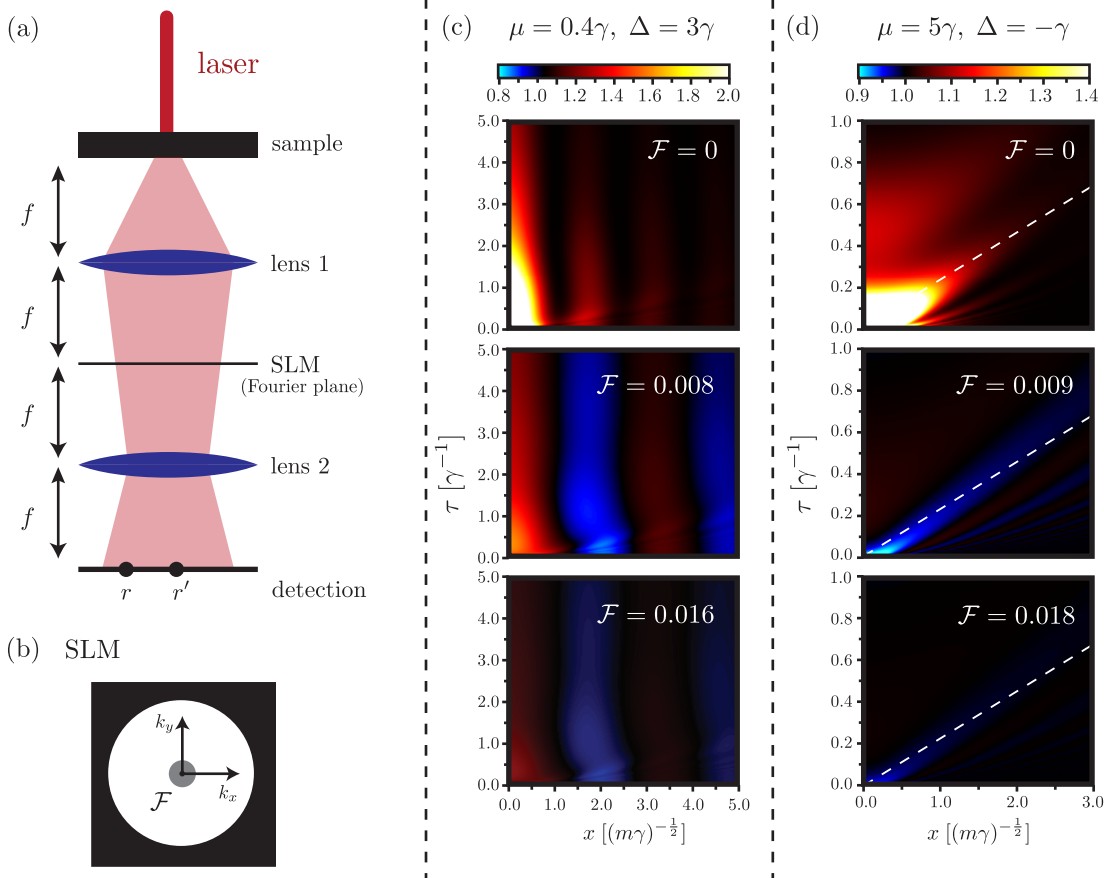

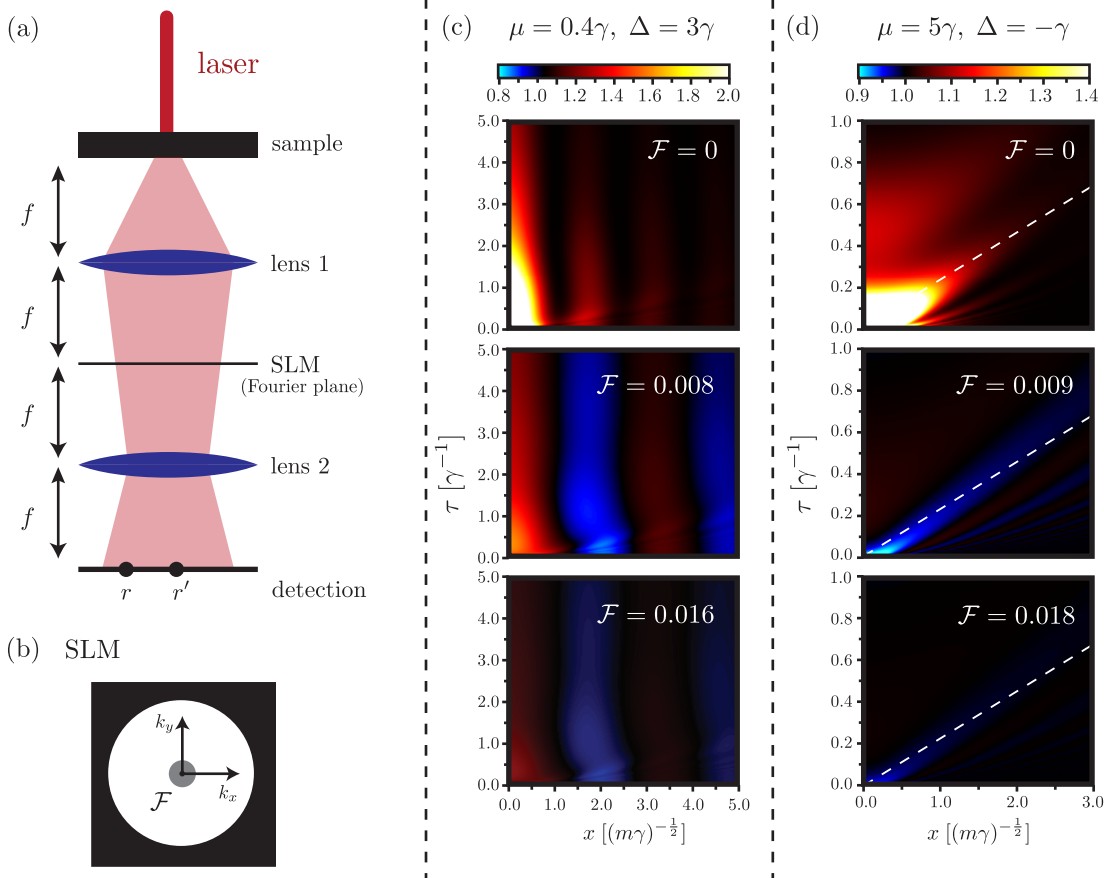

Figure 4: (a) The confocal two-lens setup to measure density-density correlations of a fluid of light. A spatial light modulator (SLM) is placed in the common Fourier plane of the two lenses to perform the desired **k**-space selection. Delayed correlations between photon detections separated by a time interval $\tau$ allow to measure the $g^{(2)}$ intensity correlation function defined in (35). (b) The spatial profile of the SLM that is used for the shaping of the beam. All modes in the 2D plane are transmitted, except a small disk centered around $\mathbf{k} = 0$, where the coherent field is situated. White corresponds to transmission, black to full blocking and gray to attenuating with a factor $\mathcal{F}$, in order to transmit a coherent field $\psi_f = \mathcal{F}\psi_0$. (c-d) Spatial-temporal profiles of the density-density correlation function $g^{(2)}(x, \tau)$ for varying (top to bottom) filtering fraction $\mathcal{F}$ (see (36)) and different (left/right) pumping parameters. Red shades correspond to bunching and blue to antibunching. For the parameters of (c), the parametrically amplified modes give rise to a spatial pattern of alternating bunching and antibunching, which turns into complete bunching for $\mathcal{F} \to 0$. The quasiparticle dispersion of this mean-field configuration is plotted in Fig. 2(c). For the parameters of (d) We notice the appearance of an approximate sound cone $x = \tau/2c$ (white dashed lines) of antibunched correlations. Also here, when $\mathcal{F} \to 0$, the antibunching turns into bunching. The quasiparticle dispersion for this case is plotted in Fig. 2(a).

$(2\mu < \gamma)$, the homogeneous state remains dynamically stable and quasiparticles generated by quantum fluctuations eventually decay. In the most interesting case $2\mu < \Delta$, shown in Fig. 4c), the set of diffusive modes (indicated by a blue band) has a disk shape in $\mathbf{k}$ space around a nonzero momentum. Since the lifetime $\tau_k = 1/(\gamma - 2\Gamma_k)$ (with $\Gamma_k = \mathfrak{J}\omega_k$) of these modes can be substantially longer than non-amplified modes for which $\tau_k = 1/\gamma$, we expect that they may leave an important imprint on the photon statistics.

In Fig. 4c), we show the spatial-temporal profiles of $g^{(2)}(x, t)$ for a fluid pumped with mean-field parameters $\mu = 0.4\gamma$ and $\Delta = 3\gamma$ and for varying displacement field $\psi_f$, which we engineer again from the condensate mode with the SLM (see Fig. (a-b)). For these parameters we have a disk of diffusive-like modes centred around nonzero momentum $k_c \approx 2.3\sqrt{m\gamma}$ (see the corresponding spectrum in Fig. 2c)). Due the parametric amplification of these modes, we see that a standing-wave-like pattern appears in $g^{(2)}(x, t)$, with a wavelength corresponding to the parametrically amplified wave vectors. Remarkably, the vanishing real part of the frequency of these modes implies a zero group velocity $v_k^g = \partial_k\omega_k$, so that the spatial pattern persists in time, practically without moving.

Upon varying $\mathcal{F}$, we can again switch from a profile with alternating bunching and anti-bunching regions in space (lower panels of Fig. 4c, with $\mathcal{F} = 0.008, 0.016$) to a profile with only bunching when the condensate is attenuated completely (panel with $\mathcal{F} = 0$). There is an optimum at about $\mathcal{F} \approx 0.008$, which stabilizes a temporal band with minimum density correlations $g^{(2)}(x, t)$ at a separation of $x \approx 2(m\gamma)^{-1/2}$. In all cases, we see that the spatial structure, as imprinted by the parametrically amplified modes, is well preserved in time. The temporal duration of the interesting correlations is now substantially longer, thereby facilitating measurement with realistic photo-detectors with nonzero photon collection time. On the other hand, the strong suppression of $g^{(2)}(x, t)$ at nonzero $x$ cannot be used to generate strongly antibunched photon statistics, since it relates to correlations between two spatially separated points.

Another interesting feature of Fig. 4c) is the presence, at short times, of small ripples on top of the otherwise very stable space-time structure, which then quickly propagate away and disappear. As for the additional features that were visible on top of the lightcones in Fig. 4d), we attribute these features to the presence of modes with a nonzero group velocity. Just outside the parametrically amplified disk, we can even verify that the modes exhibit a diverging group velocity, as can be seen on Fig. 2(b-c).

## 5 Conclusion

In this work we have theoretically investigated the peculiar nature of quantum fluctuations displayed by the light field in nonlinear planar microcavities. The fluctuations are generated as a result of pair-creation processes of quasiparticles with nonzero momentum and we have proposed free-space optics configuration to manipulate their shape and intensity.

In the first part of this work, we have shown how an appropriately designed selection and interference scheme allows to translate the intrinsic two-mode squeezing due to optical nonlinearities into a single-mode squeezed output state with strongly antibunched photon statistics. Even though our results can be placed within the framework of the Unconventional Photon Blockade [15–18], the planar microcavity geometry differs from the usual two-cavity geometry typically considered in this literature.

In the second part we have illustrated how the spatial-temporal structure of the density-density correlation function can be reconstructed and shaped with free-space optics. By filtering the beam in the far-field, as to attenuate the $k = 0$ coherent component, the nontrivial spatial-temporal profile of the correlation function can be manipulated. Upon removing

the condensate mode completely, the statistics generated by the fluctuations only leads to a strongly bunched signal. When we interfere an attenuated coherent mode with the fluctuations, we reconstruct a reinforced copy of the spatial profile of density-density correlations inside the fluid of light. In this framework, we have first discussed the high-density regime, where the correlation function exhibits an antibunching features at the coincidence point and then a lightcone-like behavior away from it. We then showed that the presence of a set of parametrically amplified modes with nonzero momentum leads to an oscillating spatial pattern that is well preserved in time.

From a wide perspective, the results of this manuscript suggest a new avenue to generate interesting quantum states of light using planar microcavity devices displaying only a weak nonlinearity.

# Acknowledgements

**Funding information**  MVR gratefully acknowledges support in the form of a Ph. D. fellowship of the Research Foundation - Flanders (FWO) and hospitality at the BEC Center in Trento. MW acknowledge financial support from the FWO-Odysseus program. IC was funded by the EU-FET Proactive grant AQuS, Project No. 640800, and by Provincia Autonoma di Trento, partially through the project "On silicon chip quantum optics for quantum computing and secure communications (SiQuro)". AI was funded by an ERC Advanced investigator grant (POLTDES). SR acknowledges support from the ETH Zürich Postdoctoral Fellowship Program.

# A  The Bogoliubov transformation revisited

The Bogoliubov approximation provides an approximate description of a quantum field in terms of a coherent condensate with Gaussian quantum fluctuations on top. Although the formalism for a driven-dissipative fluid is very similar to the equilibrium case, we would like to draw the attention to a few notable differences.

For equilibrium atomic condensates the quantity $\omega_k$ from (12) is generally known as the quasiparticle spectrum and it indicates the frequency at which a particular Bogoliubov mode $\hat{\chi}_{\mathbf{k}}$ oscillates [29]. Caution must be taken when this view is generalized to out of equilibrium systems. The reason is that, in contrast with an equilibrium condensate, the bare-particle dispersion $\varepsilon_{\mathbf{k}}$ of non-equilibrium ones is not necessarily a positive function of $\mathbf{k}$. In a driven-dissipative quantum fluid the condensate phase is set by the detuning $\delta$, a tunable parameter in experiment, while at equilibrium it is fixed by the chemical potential $\mu$, such that in that case the gapless phonon condition $\Delta = 0$ holds exactly (see (9)).

## A.1  Evolution of the quasiparticle operators

Following expression (10), the evolution of the particle operators $\hat{\phi}_{\mathbf{k}}$ can be formulated as

$$i\partial_t \begin{pmatrix} \hat{\phi}_{\mathbf{k}} \\ \hat{\phi}^{\dagger}_{-\mathbf{k}} \end{pmatrix} = \left( B_{\mathbf{k}} - i\frac{\gamma}{2} \right) \begin{pmatrix} \hat{\phi}_{\mathbf{k}} \\ \hat{\phi}^{\dagger}_{-\mathbf{k}} \end{pmatrix} + \begin{pmatrix} \hat{\xi}_{\mathbf{k}} \\ \hat{\xi}^{\dagger}_{-\mathbf{k}} \end{pmatrix}, \quad B_{\mathbf{k}} = \begin{pmatrix} \varepsilon_{\mathbf{k}} + \mu & \mu \\ -\mu & -\varepsilon_{\mathbf{k}} - \mu \end{pmatrix}. \quad (37)$$

We can always write $B_{\mathbf{k}} = U_{\mathbf{k}} D_{\mathbf{k}} U_{\mathbf{k}}^{-1}$ with

$$D_{\mathbf{k}} = \begin{pmatrix} \omega_{\mathbf{k}} & 0 \\ 0 & -\omega_{\mathbf{k}} \end{pmatrix}, \quad U_{\mathbf{k}} = \begin{pmatrix} v_{1,\mathbf{k}}^{(+)} & v_{1,\mathbf{k}}^{(-)} \\ v_{2,\mathbf{k}}^{(+)} & v_{2,\mathbf{k}}^{(-)} \end{pmatrix}, \quad v_{1,\mathbf{k}}^{(\pm)} = -\frac{\mu}{\varepsilon_{\mathbf{k}} + \mu \pm \omega_{\mathbf{k}}} v_{2,\mathbf{k}}^{(\pm)} \quad (38)$$

such that $V_{\mathbf{k}}^{(\pm)} = \left(v_{1,\mathbf{k}}^{(\pm)}, v_{2,\mathbf{k}}^{(\pm)}\right)^T$ are the left eigenvectors of $B_{\mathbf{k}}$ with eigenvalues $\pm\omega_{\mathbf{k}}$. Notice that $B_{\mathbf{k}}$ is in general not Hermitian and that therefore left and right eigenvectors do not necessarily coincide, nor must they form an orthogonal basis.

Without any loss of generality, we can now define new quasiparticle operators $\Xi_{\mathbf{k}} = U_{\mathbf{k}}^{-1}\Phi_{\mathbf{k}}$, with $\Xi_{\mathbf{k}} = (\hat{\chi}_{\mathbf{k}}^{(+)}, \hat{\chi}_{-\mathbf{k}}^{(-)})^T$ and $\Phi_{\mathbf{k}} = (\hat{\phi}_{\mathbf{k}}, \hat{\phi}_{-\mathbf{k}}^\dagger)^T$, which evolve in time as

$$\partial_t \begin{pmatrix} \hat{\chi}_{\mathbf{k}}^{(+)} \\ \hat{\chi}_{-\mathbf{k}}^{(-)} \end{pmatrix} = \left(D_{\mathbf{k}} - i\frac{\gamma}{2}\right) \begin{pmatrix} \hat{\chi}_{\mathbf{k}}^{(+)} \\ \hat{\chi}_{\mathbf{k}}^{(-)} \end{pmatrix} + \text{noise} \tag{39}$$

## A.2 Regular modes

If we want the $\hat{\chi}_{\mathbf{k}}^{(\pm)}$ to be operators that satisfy bosonic commutation relations, they need to fulfil two conditions. First of all they must be each others Hermitian conjugate, which leads to

$$v_{1,\mathbf{k}}^{(+)} = \left(v_{2,\mathbf{k}}^{(-)}\right)^* \quad \text{and} \quad v_{2,\mathbf{k}}^{(+)} = \left(v_{1,\mathbf{k}}^{(-)}\right)^* \tag{40}$$

such that we can choose to write

$$U_{\mathbf{k}} = \begin{pmatrix} u_{\mathbf{k}} & v_{\mathbf{k}}^* \\ v_{\mathbf{k}} & u_{\mathbf{k}}^* \end{pmatrix} \tag{41}$$

Secondly, the bosonic commutation relation $[\hat{\chi}_{\mathbf{k}}^{(+)}, \hat{\chi}_{-\mathbf{k}}^{(-)}] = 1$ tells us that

$$|u_{\mathbf{k}}|^2 - |v_{\mathbf{k}}|^2 = 1 \tag{42}$$

After evaluation and making use of (38), the parameters $u_k$ and $v_k$ are found as,

$$u_k, v_k = \pm\sqrt{\frac{\varepsilon_k + \mu}{2\omega_k} \pm \frac{1}{2}} \tag{43}$$

At this point we have derived the standard text book definition of the Bogoliubov transformation without making any assumptions other than the Bogoliubov operators being bosonic [29].

## A.3 Diffusive-like modes

Diffusive-like modes are characterized by having $\varepsilon_k < 0$, such that $\omega_k$ purely imaginary and we can write $\omega_k = i\Gamma_k$ with $\Gamma_k$ real. By making use of relation (38) we can now evaluate

$$|u_{\mathbf{k}}|^2 - |v_{\mathbf{k}}|^2 = |u_{\mathbf{k}}|^2\left(1 - \frac{(\epsilon_{\mathbf{k}} + \mu)^2 + \Gamma_{\mathbf{k}}^2}{\mu^2}\right) = |u_{\mathbf{k}}|^2\left(1 - \frac{(\epsilon_{\mathbf{k}} + \mu)^2 - \varepsilon_{\mathbf{k}}(\varepsilon_{\mathbf{k}} + 2\mu)}{\mu^2}\right) = 0 \tag{44}$$

This clearly contradicts (42), meaning that either condition (40) or condition (42) cannot be satisfied. Henceforth, the Bogoliubov operators $\hat{\chi}_{\mathbf{k}}^{(\pm)}$ of diffusive-like modes cannot be bosonic.

We can choose another parametrization (not unique) for the transformation matrix $U_{\mathbf{k}}$ from (38)

$$U_{\mathbf{k}} = \begin{pmatrix} r_{\mathbf{k}} & r_{\mathbf{k}}^* \\ s_{\mathbf{k}} & s_{\mathbf{k}}^* \end{pmatrix}, \quad s_{\mathbf{k}} = \frac{-\mu}{\varepsilon_{\mathbf{k}} + \mu + i\Gamma_{\mathbf{k}}} r_{\mathbf{k}}, \tag{45}$$

where we choose the normalization $s_{\mathbf{k}} r_{\mathbf{k}}^* - r_{\mathbf{k}} s_{\mathbf{k}}^* = i$, such that

$$U_{\mathbf{k}}^{-1} = i\begin{pmatrix} s_{\mathbf{k}}^* & -r_{\mathbf{k}}^* \\ -s_{\mathbf{k}} & r_{\mathbf{k}} \end{pmatrix}. \tag{46}$$

With this choice of parametrization we derive after straightforward algebra from (38) that

$$s_k = \sqrt{\frac{\mu}{2\Gamma_k}}, \quad r_k = \frac{-\mu}{\epsilon_k + \mu + i\Gamma_k}\sqrt{\frac{\mu}{2\Gamma_k}}, \tag{47}$$

# B   The time evolution of the particle operators

We focus on the linear part of time evolution of an excitation. In general, we find that equation (10) can be solved by introducing time-dependent operators of the form

$$\hat{\phi}_{\mathbf{k}}(t) = e^{-\gamma t/2}\Big(\eta_k(t)\hat{\phi}_{\mathbf{k}}(0) + \zeta_k(t)\hat{\phi}^{\dagger}_{-\mathbf{k}}(0)\Big) + \text{noise} \tag{48}$$

Here, $\eta_k(t)$ and $\zeta_k(t)$ are time-dependent Bogoliubov coefficients. When $\Delta$ is tunable, we can distinguish three different regimes that are separated in momentum space [25, 32].

- $\Delta < 0$: $\varepsilon_k$ is positive for any $\mathbf{k}$ and the quasiparticle spectrum $\omega_k$ has a gap given by $\sqrt{|\Delta|(|\Delta| + 2\mu)}$. In the limiting case $\Delta \to 0$ the gap closes and we retrieve the familiar linear spectrum of an equilibrium condensate. Particle-like (hole-like) excitations oscillate with a frequency $\omega_k$ ($-\omega_k$), but are damped by $\gamma$, reflecting the overall finite lifetime of particles. The time-dependent Bogoliubov transformation is then found as

$$\eta_k(t) = |u_k|^2 e^{-i\omega_k t} - |v_k|^2 e^{i\omega_k t}, \quad \zeta_k(t) = 2iu_k v_k^* \sin(\omega_k t). \tag{49}$$

- $0 < \Delta < 2\mu$: A disk of modes $k < \sqrt{2m\Delta}$ appears where $\varepsilon_k < 0$. Modes in this region have a purely imaginary frequency $\omega_k = i|\omega_k|$ so that they are damped or amplified at a rate $\Gamma_k = |\omega_k|$. Therefore one branch of excitations will be strongly damped in time with a lifetime $1/(\gamma + 2\Gamma_k)$, while excitations on the other branch may have a much longer lifetime $1/(\gamma - 2\Gamma_k)$ and are parametrically amplified. Modes in this region are traditionally called *diffusive*-like [25]. We derive that their time evolution is governed by

$$\eta_k(t) = i\Big(s_k r_k^* e^{\Gamma_k t} - s_k^* r_k e^{-\Gamma_k t}\Big), \quad \zeta_k(t) = -2i|s_k|^2 \sinh(\Gamma_k t), \tag{50}$$

with $s_k$ and $r_k$ given in (47). Note that, in order to have only exponentially damped diffusive modes, we must ensure that $\gamma > 2\mu$.

- $2\mu < \Delta$: Same as above, but in this case the diffusive modes are found on a ring $\sqrt{2m(\Delta - 2\mu)} < k < \sqrt{2m\Delta}$, while modes in the inner disk $k < \sqrt{2m(\Delta - 2\mu)}$ oscillate with a real frequency $\omega_k$, like usual. For these modes the time evolution is found as

$$\eta_k(t) = |u_k|^2 e^{i\omega_k t} - |v_k|^2 e^{-i\omega_k t}, \quad \zeta_k(t) = -2iu_k v_k^* \sin(\omega_k t), \tag{51}$$

with $u_k$ and $v_k$ given in (43).

# C   Main noise sources

In this Appendix, we discuss the influence of the dominant noise sources in the setup. First, we assume a homogeneous distribution of polaritons in the plane of the microcavity; this can be distorted in the presence of cavity disorder. Second, interaction with a phonon bath may lead to pure dephasing of polaritons, thus altering the squeezing properties of the light. Eventually, this together with other relaxation mechanisms may result in a population of thermal polaritons in the microcavity.

## C.1   Disorder

We may give an estimate for the effects of disorder by considering the Fourier transform $V_k$ of a random potential, $V(\mathbf{r}) = \frac{1}{\sqrt{V}}\sum_{\mathbf{k}} V_{\mathbf{k}} e^{i\mathbf{k}\cdot\mathbf{r}}$, which is applied to the planar microcavity. We

then find that the random potential enters into the evolution of the mean field in the rotating frame as

$$i\dot{\psi}(\mathbf{r}) = \left(-\frac{\nabla^2}{2m} + g|\psi(\mathbf{r})|^2 + V(\mathbf{r}) - \delta - i\frac{\gamma}{2}\right)\psi(\mathbf{r}) + F. \tag{52}$$

When we restrict to evaluating the linear response of the mean field to disorder, we find that the non-uniform polariton field can be formulated as $\psi(\mathbf{r}) = \psi_0 + \frac{1}{\sqrt{V}}\sum_{\mathbf{k}}\delta\psi_{\mathbf{k}}e^{i\mathbf{k}\cdot\mathbf{r}}$. After substitution in (52) and collecting terms up to linear order in $\delta\psi_{\mathbf{k}}$ and $V_{\mathbf{k}}$, we find a set of linear equations for each mode

$$\mathcal{L}_k\begin{pmatrix} \delta\psi_{\mathbf{k}} \\ \delta\psi_{-\mathbf{k}}^* \end{pmatrix} = \begin{pmatrix} -V_{\mathbf{k}}\psi_0 \\ V_{-\mathbf{k}}\psi_0^* \end{pmatrix} \tag{53}$$

with the response matrix

$$\mathcal{L}_k = \begin{pmatrix} \epsilon_k + gn_0 - i\frac{\gamma}{2} & g\psi_0^2 \\ -g\psi_0^{*2} & -\epsilon_k - gn_0 - i\frac{\gamma}{2} \end{pmatrix}, \tag{54}$$

and $\epsilon_k = k^2/2m - \delta + Un_0$. By solving (53) we derive the response of the density distribution to the disorder potential in the linear regime

$$\delta n_{\mathbf{k}} = |\delta\psi_{\mathbf{k}}|^2 = V_{\mathbf{k}}^2 n_0 \frac{\epsilon_k^2 + \gamma^2/4}{\left(\omega_k^2 + \gamma^2/4\right)^2} \tag{55}$$

with $\omega_k$ given in (12).

For this qualitative analysis we consider $\omega_k \approx \epsilon_k$, such that $\delta n_{\mathbf{k}} \approx V_{\mathbf{k}}^2 n_0/\left(\left(\omega_k^2 + \gamma^2/4\right)\right)$. When we compare this with the momentum distribution from coherent pair creation (16), we conclude that $\langle V^2 \rangle \Delta V_c \ll g \cdot \mu/2$, where $\langle V^2 \rangle$ is the variation of the disorder potential and $\Delta V_c$ is the correlation volume of disorder. Plugging in numbers, we find that $V(\mathbf{r})$ should not vary more than $\sim 40\mu eV$ over a scale of $1\mu m$ for an interaction constant $g = 10\mu eV \cdot \mu m^2$, in line with [14], and $\mu = 300\mu eV$. Probably, this is the primary challenge to implement our proposal in experiment, based on values reported in Ref. [38].

## C.2 Pure dephasing

Pure dephasing arises when the polariton fluids interacts with a thermal bath of phonons, present in the material. In the Markov approximation, we find that this amounts to including jump operators of the form

$$\Delta V_c \Psi^\dagger(\mathbf{r})\Psi(\mathbf{r}) \approx \Delta V_c\left(n_0 + \sqrt{\frac{n_0}{V}}\sum_{\mathbf{k},\mathbf{k}'} e^{i(\mathbf{k}-\mathbf{k}')\cdot\mathbf{r}}\left(\hat{\phi}_{\mathbf{k}} + \hat{\phi}_{\mathbf{k}'}^\dagger\right)\right), \tag{56}$$

where $\Delta V_c$ is the correlation volume of the phonons, which we estimate for simplicity as $\Delta V_c = \lambda_{\mathrm{dB}}^2$, the de Broglie wavelength of phonons. Notice that this may be somehow more complicated when the full functional form of the phonon distribution is considered. Upon explicitly evaluating the Lindblad equation with dissipators of form (56) and integrating over space, we find that

$$\partial_t n_{\mathbf{k}}\Big|_{\mathrm{deph}} = \gamma_{\mathrm{deph}} n_0 \lambda_{\mathrm{dB}}^2 \tag{57}$$

Therefore, the dominant effect of dephasing will be a scattering with phonons of polaritons from the condensate, thereby ending up in nonzero momentum modes. This is quantified by

a (Markovian) rate $\gamma_{\text{deph}}$. Notice that long-wavelength phonons may not allow for a simplified Markovian treatment. Still, it is expected that (56), describing a build-up of incoherent polaritons from scattering out of the condensate, will be the main influence of pure dephasing.

The finite-momentum density resulting from these spurious scattering processes builds up linearly in time and is damped with photon decay rate $\gamma$ of losses from the cavity. Therefore, we obtain in the long time limit that $\delta n_{\mathbf{k}}\big|_{\text{deph}} = \frac{\gamma_{\text{deph}}}{\gamma} n_0 \lambda_{\text{dB}}^2$ (we take that the coherent density 16 is a number of order 1). Here, $n_0 \lambda_{\text{dB}}^2$ is the number of condensate particles within a volume $\lambda_{\text{dB}}^2$ and can be a relatively large number for a typical residual temperature of the order of $\mu = g n_0$, the chemical potential. As such, if $\gamma_{\text{deph}}$ is too large, but still $\gamma_{\text{deph}} < \gamma$, we propose to employ a pulsed excitation scheme to circumvent this issue. Given the complexity of the dephasing process, it is difficult to obtain an accurate estimate of $\gamma_{deph}$ or to extract it from the literature. Still, it is clear that the condition $\gamma_{\text{deph}} < \gamma/(n_0 \lambda_{\text{dB}}^2)$ must be satisfied for a CW pumping scheme to be employed.

### C.3 Other noise sources

As a consequence of other spurious relaxation processes, an incoherent population of polaritons can build up at the bottom of the excitation branch. This may result in an extra density of polaritons $n_{\text{inc}}$ at nonzero $k$, not generated by coherent pair-creation, which is to be added to $n_k$ in (16) and therefore reduces the squeezing of the output light. One way to circumvent this problem is also to employ a pulsed excitation scheme. Then, the polariton population is expected to build up on the time scale of the polariton lifetime, while thermalization into the bottom of the lower polariton branch requires some relaxation process, which typically occurs on a much longer time scale.

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
