# Peer review of "Engineering Gaussian states of light from a planar microcavity"

_SciPost Physics, doi:SciPost Phys. 5, 013 (2018)_

## Round 1 · Referee Report · Anonymous (Referee 4) · 2018-1-29

Strengths

1. The work, through a rigorous theoretical investigation, proposes an experimental scheme by which nonclassical states of polaritons in semiconductor microcavities could be demonstrated. An experimental proof that microcavity polaritons can host nonclassical states has been eluding experimental efforts for over 25 years. This proposal may indicate a promising route towards this goal.
2. The work is written with great clarity and impeccable style.

Weaknesses

1. Some of the most common effects preventing nonclassical states of polaritons (such as e.g. pure dephasing and additional thermal population) are not discussed in this work. The real feasibility of the experiment remains therefore questionable.
2. Citations to highly relevant works are lacking.

Report

This work presents a theoretical analysis of a spontaneous four-wave mixing process - otherwise understood as quantum fluctuations of a coherently driven polariton condensate - showing that under an appropriate collection and interference scheme for the emitted light, subpoissonian statistics could be observed even in presence of a very weak polariton nonlinearity. The phenomenon is linked to the Unconventional Photon Blockade, already described in the literature, and the analysis is carried out in terms of optimal squeezing.

While the theoretical investigation is technically correct in my opinion, and very accurate, I still think that the feasibility of the proposed experimental scheme is not thoroughly assessed by the Authors. Microcavity polaritons always present with some extrinsic effects that are almost unavoidable. Disorder is one, and in the present work it is quickly dismissed by a couple of sentences on page 14, while on page 11 it is clearly stated that the proposed phenomenon relies on the k-> -k symmetry of the system. I am wondering if an analysis in terms of small k-broadening of the emission could be carried out without too much hassle, knowing the typical k-broadening of good quality microcavities. In any case, I think that the disorder issue should be discussed in a more insightful way.

My main concern however, is not disorder but rather phenomena like pure dephasing and additional thermal (i.e. incoherent) occupation. In a planar microcavity, polaritons are prone to all sorts of scattering mechanisms, and it is known that a combination of polariton-phonon and polariton-polariton interaction can lead to incoherent occupation of lower lying modes even if the main driving field is resonant with the lowest lying mode (not to speak of the cases with positive detuning considered by the Authors). It appears from Figure 3 that the ideal condition g2(0)=8*sqrt(n_th) is not realized for sizeable values of n_th, for which instead g2(0) seems to grow much more than what expected by this ideal condition. Given the scattering with acoustic phonons at finite temperature, and the phonon induced relaxation, one always expects a very small incoherent background occupation. This occupation is hardly going to be lower than one polariton per mode, which seems much higher than the value n_th<<0.1 which seems to be required from Figure 3. Can the Authors estimate an additional incoherent occupation at finite temperature? Can they refer to previous experiments where it has been reasonably shown that under similar conditions a very small incoherent polariton occupation is produced? The same criticism holds for pure dephasing. In the original UPB proposal, it is shown that the pure dephasing rate must be much smaller than the nonlinear energy per photon, in order for UPB to survive. Acoustic-phonon scattering rates are known as a function of temperature for polaritons, and they should easily lead to an estimate of the pure dephasing rate? Is this low enough at typical temperatures?

Another important point is the fact that few relevant citations are missing from the bibliography. First - but this is not the Authors' fault as the preprint appeared after they submitted theirs - there is now a clear cut experimental demonstration of UPB in circuit QED. See arXiv:1801.04227. Second, a review article on UPB was published recently in PRA 96, 053810 (2017). It is important in my opinion to cite this paper in particular, as it already introduces the idea of optimal squeezing through interference of the output with the driving field, which is therefore not fully original. Finally, there is at least one published experiment where phenomena strongly related to optimal squeezing and UPB have been investigated for microcavity polaritons. This work (and the related theoretical proposal) should also in my opinion be cited as highly relevant to the present work.

As a minor point, figures are not ordered in the same way as they are cited in the text, which is a bit confusing.

Requested changes

1. Carefully assess the feasibility of the proposed experiment when in presence of both disorder-induced k-broadening, pure dephasing, and extrinsic additional incoherent occupation, using realistic parameters for microcavity polaritons, to the best of the Authors knowledge.
2. Complete the bibliography with the citations described in the main report.
3. Fix the order of the figures.

  • validity: ok
  • significance: good
  • originality: good
  • clarity: high
  • formatting: excellent
  • grammar: excellent

Author:  Mathias Van Regemortel  on 2018-04-16  [id 242]

(in reply to Report 1 on 2018-01-29)

1) We have added an extra appendix (C) where we discuss the primary sources of noise expected to influence the scheme, with appropriate references in the main text. In particular, we analyze the impact of a disorder potential and pure polariton dephasing and refer to numbers found in literature where possible. We also suggest that the building up of a thermal polariton population can be circumvented by using a pulsed excitation scheme. We would like to thank the referee for drawing our attention to these points.

2) The citations were added to the main text.

3) The order of the figures has been changed.

---

## Round 1 · Referee Report · Anonymous (Referee 3) · 2018-2-6

Strengths

1- The writing is impeccable.
2- The mathematical derivations are clear and well detailed.
3- The scheme to manipulate and observe non-classical photon statistic in this system is very interesting and clever.

Weaknesses

1- A more complete analysis of the scheme robustness in realistic setups is missing.
2- The motivation for engineering antibunched output Gaussian fields and why doing it with this particular setup is not stated clearly enough.

Report

This manuscript describes how to obtain antibunched output optical field from a driven nonlinear planar micro cavity. The source of anti bunching is the weak nonlinear interaction between intracavity polaritons which effectively acts as a non-degenerate parametric amplifier (NDPA) for opposite k-modes. An interference scheme of the output field and the attenuation of the k=0 driven mode allows to manipulate and measure the level of anti-bunching of a finite-k single mode output mode. The same setup also allows to reconstruct the spatial profil of the intensity correlation of the intra-cavity optical field.

I personally enjoyed reading the manuscript as it is clearly written and great attention as been devoted to the mathematical details. The scheme is also an interesting and clear example of the ``unconventional photon blockade'' as it emphasizes on the two main ingredients responsible of the phenomenon, i.e. squeezing and displacement of the optical field, in a totally different setup than the two-cavity system where it was initially introduced. The independent control of the squeezing and the field displacement via the detuning of the drive and the spatial attenuation of the k=0 mode, respectively, is in my opinion a very clever idea. The freedom offered by the spatial extend of the system (in contrast to zero-dimension cavity NDPA) adds good value to this system.

That being said, I still have two major concerns about this work. The first one goes in the same direction of the first referee (thanks to the open peer review). As the antibunching is very sensitive to any imperfections, as capture by the effective thermal population of the resulting gaussian output optical field, I think it would be important to have a short section that roughly estimates the effects of the main noise sources in such nonlinear planar micro cavities. The goal is to have an idea of the robustness of these nonclassical signatures in state-of-the-art setups.

The second concern is more about the motivation of this project itself. As much as I think the idea is clever and interesting, I personally always had some doubts on the relevance of this ``unconventional photon blockade''. In the case of standard photon blockade, anti bunching is a consequence of having a single photon state. Single photon sources and non-Gaussian states are very useful for all sort of quantum information processing schemes. However, the antibunching resulting form the ``unconventional photon blockade'' only means that the probability of having two photons is decreased, but does not ensure single-photon light field as higher-excitation states still contribute. The consequence is that the output field is still Gaussian, which takes away a lot of its utility for quantum information processing. This project is still interesting and fits very well in the body of literature concerning this phenomenon, but I would like to see a better description of the motivation to chase this nonclassical statistic in this context.

In the same line of thoughts, what makes this system more attractive than the two-cavity setup initially used in the literature or even than a DPA; the simplest system where you can observe UPB? In the conclusion, it is written: ... the planar microcavity geometry differs from the usual two-cavity geometry typically considered in this literature. I agree about this statement, but it should be clearly stated why. I also think that it is an important point (especially compared to a DPA) that should be emphasized right from the start in the introduction.

On a slightly more technical level, I think the condition to reach the bistable regime introduced in section 2.2 (from Eq. 8) is incomplete. While having a large detuning (\delta > \sqrt(3)\gamma/2) is necessary, one should also have a condition on the drive strength F>??. Unless I'm missing something, no matter the detuning, if my drive is infinitively small, I should not reach the bistable regime.

On a lighter note, there are small additional corrections:

1. After Eq. 13, the operator Chi is not defined (in the main text).
2. The phases phi_+/- introduced in Eq. 27 should be defined right after the Equation.
3. The first sentence of section 4.3 should go: ... analyze the Delta > 0 case.

Despite my concerns and comments, I think this work is of great quality and very interesting. Once the points noted above are addressed, I strongly support its publication.

Requested changes

1- Add a section on imperfection sources in actual experimental setups and a rough estimation of their effects on the antibunching.

2- Describe the motivation for engineering Gaussian output fields that exhibit antibunching.

3- Emphasize more the differences and strength of this particular system compare to the DPA and the two-cavity setup.

3- State the condition on the drive strength to reach the bistable regime.

4- Address the small corrections noted in the report.

  • validity: high
  • significance: good
  • originality: high
  • clarity: top
  • formatting: excellent
  • grammar: perfect

Author:  Mathias Van Regemortel  on 2018-04-16  [id 243]

(in reply to Report 2 on 2018-02-06)

1) We have added an extra appendix (C) where we discuss the primary sources of noise expected to influence the scheme, with appropriate references in the main text. In particular, we analyze the impact of a disorder potential and pure polariton dephasing and refer to numbers found in literature where possible. We also suggest that the building up of a thermal polariton population can be circumvented by using a pulsed excitation scheme. We would like to thank the referee for drawing our attention to these points.

2-3) The referee is perfectly right when he says that the resulting output field is still Gaussian and therefore no single photon (Fock) state. However, we give some more motivation in the introduction to explain that even this strong nonclassical feature can be useful. Moreover, we emphasize that our scheme has a larger flexibility compared to previous two-cavity models, as the squeezing and interference is now spatially separated. This was a very useful comment and addressing it has certainly improved our manuscript.

4) We mean that there is bistable behavior in function of the drive strength F, so that automatically implies that it only occurs for a range of values F. When the drive strength F is increased, at some point there is a sudden jump in polariton density.

5) We have corrected the small mistakes.

---

## Round 2 · Referee Report · Anonymous · 2018-4-20

Strengths

1. All remarks from my previous report have been addressed satisfactorily, except one.

Weaknesses

1. The remark on pure dephasing has not been addressed satisfactorily

Report

I acknowledge that the Authors have addressed all my original points. I think however that the criticism on the effect of pure dephasing has not been satisfactorily addressed. In Section C.2 the Authors compare the additional broadening induced by pure dephasing to the overall broadening, concluding that the former is negligible in typical cases. This is however not correct for unconventional blockade. In several of the original works, including Ref. 20 of the current manuscript, it is shown that UPB is destroyed when the pure dephasing rate becomes comparable to the nonlinear Kerr energy (which in the present case should be g*n_0^2). This is the result that one obtains when modeling the driven-dissipative process leading to UPB. The Authors study the occurrence of UPB in terms of an optimal squeezing analysis, which assumes a given form of the density matrix of the system, with optimized squeezing and displacement features. The point is that, this optimal state will never be achieved as the steady state of a driven-dissipative setup if the pure dephasing rate exceeds the nonlinear energy. The Authors should carry out a simulation of the driven-dissipative process (possibly restricted to few modes, for simplicity) including typical polariton dephasing rates, and hopefully show that the proposed mechanism is robust to pure dephasing in typical experimental conditions.

I recommend that the manuscript is accepted only upon this analysis, as I am afraid that the current considerations in Section C.2 are technically inaccurate.

Requested changes

Solve the quantum master equation (within a few-mode approximation) in presence of drive, dissipation and pure dephasing, to conclusively assess the effect of pure dephasing on the proposed scheme. For this, parameters relevant to state-of-the-art polariton system should be used.

  • validity: ok
  • significance: high
  • originality: high
  • clarity: ok
  • formatting: excellent
  • grammar: excellent

Anonymous on 2018-05-24  [id 258]

(in reply to Report 1 on 2018-04-20)
Category:
reply to objection

We admit that our previous analysis of dephasing was inaccurate and we would like to thank the referee for noticing this. However, we do not see the purpose of performing a hard-core simulation of the master equation for this model, which is continuous and contains a high number of particles in the condensate mode. In the new analysis, we point out that the primary effect of dephasing would be a scattering of condensate particles with phonons to nonzero momentum modes. We obtain this by evaluating the effect of the dephasing jump operator upon the system with Lindbladian dynamics. We also estimate the impact of this spurious effect, but we were unable to find accurate measures of the dephasing rate of a polariton system in the literature. In any case, a pumped excitation scheme is a way to circumvent this problem.

We hope that the referee is now willing to accept our work for publication in SciPost with this new analysis.

---

## Round 2 · Author Response

We have adapted the manuscript according to the suggestions of both referees. In particular, a study has been added where we discuss the possible impact of various noise sources upon our proposed optical scheme.

---

## Round 2 · List of Changes

- We have added a paragraph in the introduction to motivate our work better. It is outlined why photons generated with the unconventional photon blockade can still bear interesting nonclassical features. Furthermore, we explain why the scheme that we propose may have more flexibility than other setups proposed in this context.

- We have added an Appendix (C), with references in the main text, where we discuss the effect of possible noise sources on the setup.

- The order of Figs. 1 and 2 was changed

- Some typo's where corrected

---

## Round 3 · Referee Report · Anonymous (Referee 1) · 2018-5-30

Strengths

1- My remaining criticism on pure dephasing has now been addressed extensively and satisfactorily.

Report

My remaining criticism on pure dephasing has now been addressed extensively and satisfactorily.

---

## Round 3 · Author Response

We admit that our previous analysis of dephasing was inaccurate and we would like to thank the referee for noticing this. However, we do not see the purpose of performing a hard-core simulation of the master equation for this model, which is continuous and contains a high number of particles in the condensate mode. In the new analysis, we point out that the primary effect of dephasing would be a scattering of condensate particles with phonons to nonzero momentum modes. We obtain this by evaluating the effect of the dephasing jump operator upon the system with Lindbladian dynamics. We also estimate the impact of this spurious effect, but we were unable to find accurate measures of the dephasing rate of a polariton system in the literature. In any case, a pulsed excitation scheme is a way to circumvent this problem.

We hope that the referee is now willing to accept our work for publication in SciPost with this new analysis.

---

## Round 3 · List of Changes

We have revised the discussion of pure dephasing in appendix C.2.

---

## Editorial Decision

published